# Numerical Study on Death of Squamous Cell Carcinoma Based on Various Shapes of Gold Nanoparticles Using Photothermal Therapy

**DOI:** 10.3390/s22041671

**Published:** 2022-02-21

**Authors:** Donghyuk Kim, Hyunjung Kim

**Affiliations:** Department of Mechanical Engineering, Ajou University, Suwon-si 16499, Gyeonggi-do, Korea; kimdonghyuk20@ajou.ac.kr

**Keywords:** apoptosis, continuous wave laser, gold nanoparticles, heat transfer, hyperthermia, numerical analysis, photothermal therapy, squamous cell carcinoma, thermal damage

## Abstract

Due to increased exposure to ultraviolet radiation caused by increased outdoor activities, the incidence of skin cancer is increasing. Incision is the most typical method for treating skin cancer, and various treatments that can minimize the risks of incision surgery are being investigated. Among them, photothermal therapy is garnering attention because it does not cause bleeding and affords rapid recovery. In photothermal therapy, tumor death is induced via temperature increase. In this study, a numerical study based on heat transfer theory was conducted to investigate the death of squamous cell carcinoma located in the skin layer based on various shapes of gold nanoparticles (AuNPs) used in photothermal therapy. The quantitative correlation between the conditions of various AuNPs and the laser intensity that yields the optimal photothermal treatment effect was derived using the effective apoptosis ratio. It was confirmed that optimal conditions exist for maximizing apoptosis within a tumor tissue and minimizing the thermal damage to surrounding normal tissues when using AuNPs under various conditions. Furthermore, it is envisioned that research result will be utilized as a standard for photothermal treatment in the future.

## 1. Introduction

Recently, ultraviolet exposure has increased due to increased outdoor activities as a result of global economic development. Accordingly, the incidence of skin cancer is increasing annually [1,2]. Skin cancer can be primarily classified into squamous cell carcinoma, basal cell carcinoma, and malignant melanoma, and the treatment of these skin cancers is performed using various methods such as chemotherapy, cryotherapy, and incision [3,4,5]. These therapy methods, however, pose various negative effects [6,7,8]. In particular, surgery via incision causes bleeding and possible secondary infection [9,10].

An alternative treatment (i.e., photothermal therapy) is garnering attention as a method that can alleviate these adverse effects [11,12]. Photothermal therapy is a treatment method that uses the photothermal effect, a phenomenon in which light energy is converted into thermal energy when light energy is irradiated onto a medium, to induce the death of tumor tissues by increasing the temperature of the target tumor tissue [13,14]. This treatment method affords quick recovery and minimal risk of subsequent infection [15,16]. Photothermal therapy primarily supplies heat to biological tissues via laser, thereby affording the easy control of the heating range and intensity [17,18]. When a visible ray laser is used (among lasers of various wavelengths), a significant amount of light absorption occurs not only in the target tumor tissue, but also in normal tissues. Hence, laser in the near-infrared region, which involves a low light-absorption coefficient in biological tissues, is applied in photothermal therapy [19]. However, because the tumor tissue has a low light absorption rate with respect to laser in the near-infrared region, a light absorption enhancer is injected into the tumor tissue to increase the light absorption rate of the tumor tissue to perform treatment. This causes a temperature increase in only the tumor tissue, enabling selective treatment [20]. Among the various light absorption enhancers, gold nanoparticles (AuNPs) are widely used because they are harmless to the human body and afford high surface workability [21,22,23]. AuNPs reach the tumor in various ways such as direct administration and intravenous injection, and are injected into the tumor through a process such as endocytosis and used for treatment [24,25,26].

Biological tissues including tumor tissues cause various types of death depending on temperature [27,28]. Apoptosis occurs between 43 °C and 50 °C, and is a form of self-death that does not affect the surroundings. Necrosis, in contrast, occurs at temperatures of 50 °C or above; it poses the risk of cancer cell metastasis and recurrence as it affects adjacent tissues when the tissue dies. Accordingly, the temperature range corresponding to apoptosis should be maintained to prevent necrosis. In photothermal therapy, treatment is performed by controlling the appropriate intensity of the heat source, injection amount of AuNPs, and type of AuNPs to minimize thermal damage to surrounding normal tissues while maintaining a temperature band corresponding to apoptosis [29].

Based on these factors, various studies regarding photothermal therapy are being conducted. Nam et al. [30] conducted an experimental study pertaining to photothermal therapy combined with chemotherapy. Polydopamine-coated spike-shaped AuNPs with excellent photothermal stability and optical efficiency were developed to enable photothermal therapy; consequently, successful treatment for 85% of CT26 colon carcinoma cases was confirmed. In addition, the therapeutic efficacy of TC-1 submucosa-lung metastasis was confirmed. Mackey et al. [31] conducted a study regarding photothermal therapy using rod-type AuNPs. The study was conducted both theoretically and in vitro, and information pertaining to the shapes of three sizes of gold nanorods that afforded the optimal therapeutic effect was obtained. Finally, it was demonstrated that gold nanorods with a length of 28 nm and a diameter of 8 nm were the most effective light absorption enhancers. Broek et al. [32] conducted an experimental study regarding photothermal therapy using branched AuNPs. In this study, branched AuNPs were biofunctionalized with nanobodies of heavy chain-only antibodies and bound to HER2 antigen expressed in both breast and ovarian cancer cells. As a result of the treatment, it was confirmed that the tumor died when 690 nm continuous wave (CW) laser was irradiated with an intensity of 38 W/cm^2^ for 5 min, and that the tumor was killed only under an optical density of 4 or higher when the corresponding AuNP was used. Xi et al. [33] developed new photothermal agents (PTAs) to improve the low photothermal conversion efficiency (PCE) of conventional PTAs. PCE is facilitated by the -CF3 moiety included in the meso-position of the BODIPY scaffold in the PTAs because the free rotation of -CF3 provides a pathway for efficient non-radiative decay. In addition, it was confirmed that the barrier-free rotation of -CF3 continued even when tfm-BDP was encapsulated with polymer nanoparticles. Excellent therapeutic effects were confirmed in both in vitro and in vivo experiments, and mouse experiments confirmed that tfm-BDPNP was efficiently accumulated in the tumor area and completely resected tumor tissue when near-infrared lasers of 0.3 W/cm^2^, 808 nm were used.

In summary, studies regarding photothermal therapy under various conditions have been conducted both experimentally and theoretically. However, treatment trends based on different types of AuNPs and laser irradiation conditions could not be determined, and studies suggesting optimal treatment conditions quantitatively when considering various AuNP concentrations and other conditions at the same time are insufficient. In addition, although photothermal therapy induces cell death through temperature increase due to the photothermal effect, studies pertaining to heat transfer are insufficient. Therefore, in this study, a numerical study of photothermal therapy based on heat transfer theory was conducted for actual skin layers including squamous cell carcinoma. The temperature distribution in tumor and normal tissues based on various AuNPs shapes, laser intensities, and volume fractions of injected AuNPs was obtained. In addition, the optimal treatment effect was proposed by quantitatively confirming conditions that maximize the apoptosis of tumor tissues while minimizing thermal damage to surrounding normal tissues, which is the main purpose of photothermal therapy based on the apoptotic variable proposed by Kim et al. [34].

## 2. Materials and Methods

### 2.1. Discrete Dipole Approximation (DDA) Method

In this study, the DDA method [35,36] was used to calculate the optical efficiency of various nanoparticles. In this method, the scattering and absorption characteristics are calculated after assuming that dipoles with polarized shapes are formed at regular intervals based on information regarding a specific shape. Compared with the Mie theory [37,38], which can be applied only to existing spheres or ellipses, the DDA method allows a wide range of shapes that can be calculated and applied. In addition, compared to the finite difference domain method, the DDA method has the advantage of being very fast in computation speed and requiring less memory for computation [39].

To perform calculations using the DDA method, the polarization vector *P* must be determined. *P* depends on the interaction between the dipole and local electric field *E*; it can be calculated using Equation (1), where α is the polarizability, and *r* is the position vector. In addition, the local electric field *E* can be calculated using Equation (2).
(1)$$P_{i}=\alpha_{i}\cdot E_{i}\left(r_{i}\right)$$
(2)$$E_{i}\left(r_{i}\right)=E_{inc,i}-\overset{N}{\underset{j\neq i}{\sum }}A_{ij}\cdot P_{j}\;\;\left(i,j=1,2,3,\ldots ,N\right)$$
(3)$$E_{inc,i}=E_{0}e^{i\left(k\cdot r_{i}\right)}$$
(4)$$A_{ij}\cdot P_{j}=\frac{e^{i\left(k\cdot r_{ij}\right)}}{r_{ij}^{3}}\left\{k^{2}r_{ij}\times \left(r_{ij}\times P_{j}\right)+\frac{1-ikr_{ij}}{r_{ij}^{2}}\times [k^{2}P_{j}-3r_{ij}\left(r_{ij}\cdot P_{j}\right)]\right\}\;\;\left(\;i\neq j\right)$$
where *k* is the wavenumber of the radiation. The electric field at position *i* due to the dipole at position *j* is included in the second term on the right side of Equation (2). Meanwhile, A is the interaction matrix between the dipoles (Equation (4), where $r_{ij}$ is $\left|r_{i}-r_{j}\right|$, and $A_{ij}$ is the interaction matrix under the condition that i≠j. If *i* and *j* are the same, then the interaction matrix can be simplified as $\alpha_{i}^{-1}$. Subsequently, Equation (2) can be transformed into a three-dimensional complex linear equation; hence, the *P* of each dipole can be calculated.

In the case of the absorption, attenuation, and scattering, cross-sections can be calculated as shown in Equations (5)–(7) using the calculated *P*, where * denotes complex conjugation.
(5)Cabs=4πkE02∑i=1NImPi·αi−1*Pi*−23k3PiPi*
(6)Cext=4πkE02∑i=1NImEinc,i*·Pi
(7)$$C_{sca}=C_{ext}-C_{abs}$$

Finally, the absorption, attenuation, and scattering efficiencies can be calculated using Equation (8), where $r_{eff}$ is the effective radius of the particle, and *V* is the volume of the particle.
(8)$$Q_{abs}=\frac{C_{abs}}{\pi r_{eff}^{2}},\;\;Q_{ext}=\frac{C_{ext}}{\pi r_{eff}^{2}},\;\;Q_{sca}=\frac{C_{sca}}{\pi r_{eff}^{2}}$$
(9)$$r_{eff}={\left(\frac{3V}{4\pi }\right)}^{\left(1/3\right)}$$

### 2.2. Heat Transfer Model and Optical Properties

In this study, the Pennes bioheat equation [40], which is widely used in the field of bio-heat transfer, was used for the thermal analysis of biological tissues. In this equation, it is assumed that the heat generated by blood and metabolism is uniformly generated in the biological tissue, as expressed in Equation (10).
(10)$$\rho c_{p}\frac{\partial T}{\partial t}=k_{m}\nabla^{2}T+q_{b}+q_{met,}$$
where ρ is the density; $c_{p}$ is the specific heat; T is the temperature; and $k_{m}$ is the thermal conductivity of the medium. $q_{b}$ and $q_{met}$ are the heat generated by blood flow and metabolism, respectively. In this study, the skin surface was irradiated with a Gaussian profile laser, and the temperature was increased via the photothermal effect. Therefore, the heat generated by the laser must be simultaneously considered [41]. In addition, the heat generated by metabolism and the blood flow is insignificant compared with the heat generated by the laser, and Equation (10) is written as shown in Equation (11) to confirm the study results under steady-state conditions.
(11)$$-k_{m}\nabla^{2}T=q_{l}$$
(12)$$q_{l}=\mu_{abs}\frac{P_{l}}{\pi r_{l}}e^{-\mu_{tot}z}\cdot e^{-\frac{r^{2}}{r_{l}^{2}}}\;\;\;\;\;\left(\mu_{tot}=\mu_{abs}+\mu_{sca}^{\prime }\right),$$
where $\mu_{abs}$ is the light absorption coefficient of the medium; $P_{l}$ is the laser intensity; and $r_{l}$ is the laser radius. $\mu_{tot}$ represents the total light attenuation coefficient of the medium and is the sum of the absorption coefficient ($\mu_{abs}$) and attenuated scattering coefficient ($\mu_{sca}^{\prime }$) of the medium.

The optical properties of the medium should be calculated separately for normal and tumor tissues. For the former, because AuNPs are not injected into them, only their optical properties are considered. However, for the tumor tissues, because various shapes of AuNPs are injected inside them, the corresponding optical properties must be considered simultaneously.
(13)$$\mu_{abs,n}=0.75f_{v}\frac{Q_{abs,n}}{r_{eff}},\;\;\mu_{sca,n}=0.75f_{v}\frac{Q_{sca,n}}{r_{eff}}$$
(14)$$\mu_{sca,n}^{\prime }=\mu_{sca,n}\left(1-g\right)$$
(15)$$\mu_{abs}=\mu_{abs,n}+\mu_{abs,m},\;\;\mu_{sca}^{\prime }=\mu_{sca,n}^{\prime }+\mu_{sca,m}^{\prime }$$

Equations (13)–(14) are equations that represent the optical properties of AuNPs [42], where $f_{v}$ is the volume fraction of AuNPs in the tumor; *g* is the anisotropy coefficient; and $Q_{abs}$ and $Q_{sca}$ are the absorption and scattering efficiencies of the particles, respectively. Finally, the optical properties of the medium containing AuNPs were calculated as the sum of the optical properties of the AuNPs and the medium, as shown in Equation (15). The optical property of the medium is not a calculated value, but an intrinsic property of the material. The optical efficiency of AuNPs was calculated using the DDA method described above, and it was assumed that the injected AuNPs were uniformly distributed in the medium [43].

### 2.3. Apoptotic Variables

In this study, three variables proposed by Kim et al. [34] were used to quantitatively confirm the therapeutic effect of photothermal treatment.

First, the apoptosis ratio ($\theta_{A}$), which represents the quantity of tumor tissues that undergo apoptosis at a temperature range of 43 °C–50 °C, was calculated as the ratio of the volume of the tumor tissue to the tissue volume entering the temperature band corresponding to apoptosis. For example, if all regions of the tumor are in the temperature band of apoptosis, then $\theta_{A}$ is 1.
(16)$$\theta_{A}=\frac{apoptosis\;volume\;(if\;43<V\left(T\right)<50)}{tumor\;volume}$$
(17)$$\theta_{H}=\frac{\sum_{i=1}^{n}\left(V_{i}\left(T\right)\cdot w_{i}\right)}{V_{i}}$$

Next, the thermal hazard value ($\theta_{H}$), which quantitatively represents the amount of thermal damage to normal tissues, is the ratio between the normal tissue volume and weighted sum that affects each phenomenon based on the temperature of the biological tissue, as shown in Equation (17). The phenomena that occur in each temperature range in the biological tissue are shown in Table 1 [44,45]. The minimum value of $\theta_{H}$ was 1, indicating that all regions of the normal tissue corresponded to 37 °C to 43 °C.

As shown in Equation (17), the result for $\theta_{H}$ varies depending on the range of normal tissues for calculating thermal damage. Accordingly, in this study, the range of the normal tissue surrounding the tumor tissue was selected as 50% of the length of the tumor tissue.

Finally, the effective apoptosis ratio ($\theta_{eff}$), which quantitatively represents the maximal occurrence of apoptosis in tumor tissues and the minimal occurrence of thermal damage to surrounding normal tissues simultaneously (i.e., the ultimate purpose of photothermal therapy), is the ratio between $\theta_{A}$ and $\theta_{H}$, as shown in Equation (18). Because $\theta_{eff}$ is the relative value between the ratio of the volume corresponding to the temperature band where apoptosis occurs in the tumor tissue and the amount of thermal damage to the surrounding normal tissues, more effective treatment is induced as $\theta_{eff}$ increases. Finally, the optimal conditions for photothermal therapy can be determined using $\theta_{eff}$.
(18)$$\theta_{eff}=\frac{apoptosis\;ratio\left(\theta_{A}\right)}{thermal\;hazard\;value\left(\theta_{H}\right)}$$

### 2.4. Validation of Numerical Model

To verify the numerical analysis model to be used in this study, the studies of Soni et al. [46] and Ren et al. [47] were used. The study assumed that a cylindrical tumor was located from the surface in cylindrical normal tissues as shown in Figure 1, and assumed that AuNPs in the tumor were uniformly distributed. Normal tissue has a radius of 20 mm and a depth of 10 mm, and a tumor tissue has a radius of 10 mm and a depth of 5 mm. The radius of the irradiated laser was selected to be 10 mm, which is the same as the radius of the tumor, and the intensity of the laser was 0.5 W/cm^2^. In addition, convection with air exists at the skin surface, and the initial temperature of all media was assumed to be 37 °C. The thermal and optical properties of normal and tumor tissues are summarized in Table 2.

Figure 2 is a graph of the validation results with the numerical analysis model in this study. The temperature along the radial direction when the depth was 0 mm and 5 mm in the central part of the tumor was confirmed, and the error was within about 1.4%. Through this, it was confirmed that the numerical analysis model used in this study was valid.

### 2.5. Numerical Investigation

In this study, a numerical analysis of photothermal therapy was conducted on a skin structure composed of four layers that included squamous cell carcinoma, as shown in Figure 3. The radius and depth of the entire normal tissue were 10 and 6 mm, respectively. It was assumed that tumor tissues with a radius of 5 mm and a depth of 2 mm were present on the skin surface. In addition, the irradiated laser exhibited a Gaussian distribution and had the same radius of 5 mm as the tumor tissue. The thickness of each skin layer and the thermal properties of all the media are summarized in Table 3. For the optical properties of the tumor and skin layer, the studies of Meglinski et al. [19] and Salomatina et al. [48] were referenced since values at various wavelengths were required.

Using the numerical analysis model, the thermal behavior of tumor tissues and the surrounding normal tissues for various laser intensities, volume fractions of injected AuNPs, and AuNP types were confirmed, and the conditions are shown in Table 4. The laser intensity was set from 0 to 1.2 W with 0.002 W intervals. Meanwhile, six types of AuNPs with diameters ranging from 10 to 50 nm were selected at intervals of 5 nm, and the volume fraction of injected AuNPs in the tumor was classified into four stages from 10^−3^ to 10^−6^.

As described above, six AuNPs were injected into the tumor, as shown in Figure 4. In the rod-type AuNPs, the aspect ratio was fixed at 6.67; in the shell-type AuNPs, the difference between the outer diameter and the inner diameter was fixed at 4 nm; and in the prism-type AuNPs, the aspect ratio was fixed at 1.67. Finally, for various shapes of AuNPs, their absorption and scattering efficiencies were calculated in the wavelength range of 500 to 1500 nm using the DDA method.

Finally, a numerical analysis of photothermal treatment was performed on the skin layer containing squamous cell carcinoma under the various conditions described above. For all cases, the temperature distribution of tumor tissues and the surrounding normal tissues was obtained, and the optimal treatment conditions were derived to maximize the temperature band corresponding to the apoptosis of tumor tissues while minimizing thermal damage to the surrounding normal tissues.

## 3. Results

### 3.1. Derivation of Optical Properties for Various AuNP Types

The optical efficiencies of AuNPs with various shapes were calculated using the DDA method. Figure 5 shows the absorption efficiency and scattering efficiency of the rod-type AuNPs. Calculations were performed for a wavelength range of 500 to 1500 nm, and it was confirmed that a laser wavelength that results in the optimal efficiency exists for each $r_{eff}$.

Analysis was performed on AuNPs of six different shapes, and the wavelength ($\lambda_{max}$) values with the maximum absorption efficiency at various shapes and $r_{eff}$ were recorded. The absorption and scattering efficiencies at the corresponding wavelengths are summarized in Table 5. In the case of $r_{eff}$, after the size of the particles is determined, it is calculated through Equation (9), and the size of the particles having a value as close as possible to the $r_{eff}$ range presented in this study was set. As shown in Table 5, the absorption efficiency of the rod type was generally greater than that of the other types of AuNPs. In addition, the absorbance efficiency increased with $r_{eff}$ increased generally. However, for the rod and shell types, the $r_{eff}$ with the optimal absorbance efficiency existed depending on $r_{eff}$. Finally, the temperature distribution inside the tumor and normal tissues when the volume fraction of AuNPs was changed to achieve the optimal optical efficiency conditions based on the shape of various AuNPs was obtained.

### 3.2. Temperature of Tumor and Normal Tissues for Various Conditions

Figure 6 shows the temperature distribution of the normal and tumor tissues when the laser intensity was 0.4 W and the volume fraction of injected AuNPs was 10^−3^ and 10^−6^, separately. The injected AuNPs were of the rod type—their aspect ratio and $r_{eff}$ were 6.67 and 50 nm, respectively. When the volume fraction of AuNPs in the tumor was high ($f_{v}$ = 10^−3^), the light absorption coefficient of the tumor tissue including AuNPs was high. Therefore, a significant amount of laser energy was absorbed from the medium. Accordingly, the temperature of the tumor region increased significantly, as shown in Figure 6a. In addition, the temperature of the surrounding normal tissue increased because of heat conduction from the tumor tissue. In contrast, when the volume fraction of AuNPs in the tumor was low ($f_{v}\;$= 10^−6^), the light absorption coefficient of the tumor tissue was low; therefore, even when a laser of the same intensity was used, the increase in temperature was insignificant, as shown in Figure 6b. The results confirmed that the temperature distribution of the tumor and surrounding normal tissues differed depending on the volume fraction of AuNPs in the tumor. Therefore, in this study, by obtaining the temperature distribution in the medium based on various shapes and volume fractions of AuNPs, the degree of thermal damage to the surrounding normal tissue as well as the corresponding temperature range that caused apoptosis were quantitatively confirmed.

### 3.3. Apoptosis Ratio

In photothermal therapy, laser is irradiated to the affected area to kill the tumor tissue by increasing the temperature via the photothermal effect. From a temperature perspective, the temperature band corresponding to apoptosis must be maintained to prevent necrosis; therefore, the extent to which the apoptosis temperature band is maintained must be determined quantitatively by verifying the temperature distribution in the tumor tissue.

Figure 7 shows a graph of the apoptosis ratio ($\theta_{A}$) as functions of $P_{l}$ and $f_{v}$ for the rod-type AuNPs with various $r_{eff}$. As shown in the graph, as $r_{eff}$ increased, the intensity of the laser with the maximum $\theta_{A}$ increased. This is because, when calculating the absorption coefficient, as $Q_{a}$ does not increase at the same rate as $r_{eff}$, the absorption coefficient decreases, and a higher laser intensity is required to achieve the corresponding temperature range in which apoptosis occurs. In addition, it was confirmed that as $f_{v}$ decreased, the intensity of the laser with the maximum $\theta_{A}$ increased. This is because as $f_{v}$ decreased, the light absorption coefficient of the medium decreased; as such, the amount of heat absorbed by the medium decreased. Hence, the amount of heat generated by the laser must be increased to achieve the temperature band in which apoptosis occurs. Based on Figure 5, the condition in which the apoptosis ratio becomes 1 for all cases can be obtained.

### 3.4. Thermal Hazard Value

One of the most significant goals of photothermal therapy is to maintain a temperature range of 43 °C–50 °C in the tumor, which is known to cause apoptosis. Although direct heating using laser is not performed to the surrounding normal tissues, temperature increase is inevitable due to conduction heat transfer from the tumor tissue. Hence, the death of surrounding normal tissues due to temperature increase must be minimized by verifying the temperature distribution of not only the tumor tissue, but also the surrounding normal tissues. Accordingly, in this study, the amount of thermal damage to the surrounding normal tissues was quantitatively confirmed based on the thermal hazard value ($\theta_{H}$) (Equation (17)).

Figure 8 shows $\theta_{H}$ as functions of $P_{l}$ and $f_{v}$ for sphere-type AuNPs with various $r_{eff}$. As shown in the graphs, as $f_{v}$ increased, $\theta_{H}$ at the same $P_{l}$ was high. This is because when $f_{v}$ increased, the light absorption coefficient of the medium increased and the amount of heat absorbed by the tumor tissue increased; therefore, the temperature of the surrounding normal tissue increased higher, and the amount of thermal damage increased. Meanwhile, as $r_{eff}$ increased, $\theta_{H}$ decreased. This is because as $r_{eff}$ increased, the light absorption coefficient of the medium decreased, as described in Section 3.3, which implies that the amount of heat absorbed by the medium decreased at the same laser intensity, and the temperature increased slowly.

### 3.5. Effective Apoptosis Ratio

Previously, the apoptosis ratio ($\theta_{A}$) and thermal hazard value ($\theta_{H}$) were used to define the conditions for maximizing the temperature range where apoptosis occurs in the tumor tissue and the relative ratio of thermal damage amount to the surrounding normal tissue. To confirm the distribution of the effective apoptosis ratio ($\theta_{eff}$), which combines the two variables above, calculations were performed based on the shape and size of the AuNPs as well as the volume fraction of the injected AuNPs.

Figure 9 shows $\theta_{eff}$ as functions of $P_{l}$ and $f_{v}$ for shell-type AuNPs with various $r_{eff}$. As shown, the $P_{l}$ value that results in the maximum $\theta_{eff}$ existed for each $f_{v}$, and that the $P_{l}$ with the maximum $\theta_{eff}$ decreased as $f_{v}$ increased. This is because, as described above, when $f_{v}$ increased, the light absorption coefficient of the medium increased and the amount of heat absorbed from the medium increased; therefore, the intensity of the laser required to maintain the temperature range where apoptosis occurs was decreased.

In addition, as $r_{eff}$ increased, the intensity of the laser at which $\theta_{eff}$ was maximized increased. However, for the shell-type AuNPs, as shown in Figure 9, the intensity of the laser that afforded the optimal treatment effect was not constant because $r_{eff}$, which maximized the absorption efficiency for various $r_{eff}$, existed. In addition, it was confirmed that an excessive increase in the laser intensity decreased $\theta_{eff}$ due to the increase in $\theta_{H}$. The results confirmed that certain values of laser intensity and volume fraction of injected AuNPs yielded the optimal therapeutic effect for the different shapes and sizes of AuNPs. Figure 8 shows the derivation results of $\theta_{eff}$ for each $r_{eff}$ for six types of AuNPs investigated in this study.

Based on the analysis, the volume fraction of injected AuNPs that resulted in the optimal therapeutic effect for all AuNP shapes was derived based on $f_{v}$ = 10^−6^. As shown in Figure 10, the optimal values of $r_{eff}$ and $P_{l}$ afforded the optimal therapeutic effect for various AuNP shapes. For the rod type, the absorption coefficient decreased as $r_{eff}$ increased; consequently, $P_{l}$ increased, and the optimal $\theta_{eff}$ was obtained. For the shell type, it was confirmed that the $P_{l}$ value that afforded the optimal therapeutic effect was minimized as the absorption coefficient was maximized when $r_{eff}$ was 25 nm.

For the pyramid and cube types, as $r_{eff}$ increased, the absorption coefficient exhibited similar values; however, when $r_{eff}$ was 40 nm, the absorption coefficient decreased and the optimal point of $P_{l}$ decreased. For the sphere and prism types, the calculated absorption coefficients were similar as $r_{eff}$ increased; as such, the optimum points of $P_{l}$ based on $r_{eff}$ were similar. Table 6 summarizes the $f_{v}$ and $P_{l}$ values that afforded the maximum $\theta_{eff}$ based on the shape and size of each AuNP.

## 4. Conclusions

In this study, a numerical study based on heat transfer theory was performed to investigate photothermal therapy using AuNPs on an actual skin layer containing squamous cell carcinoma. The temperature distribution inside the biological tissue was calculated using the Pennes bioheat equation, and the optical properties of the AuNPs were calculated using the discrete dipole approximation method.

The optimal treatment conditions for photothermal therapy were confirmed through the apoptosis ratio, which quantitatively confirms the volume ratio corresponding to the temperature band of apoptosis in tumors, the thermal hazard value that confirms the thermal damage of surrounding normal tissues, and the effective apoptosis ratio that considers the above two variables at the same time.

Finally, the AuNP shape, $r_{eff}$, $f_{v}$, $\lambda_{max}$, and $P_{l}$ that afforded the optimal therapeutic effect were obtained; hence, they can be utilized as the optimal treatment conditions for performing photothermal treatment in the future. In addition, future studies will not simply assume a cylindrical tumor, but rather conduct studies on tumors of various shapes.

## Figures and Tables

**Figure 1 sensors-22-01671-f001:**
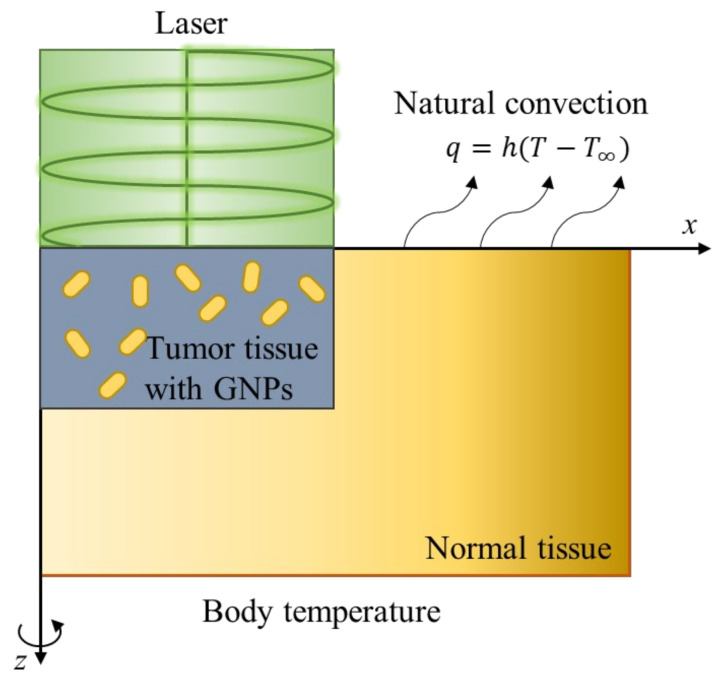
Schematic figure of the validation numerical model.

**Figure 2 sensors-22-01671-f002:**
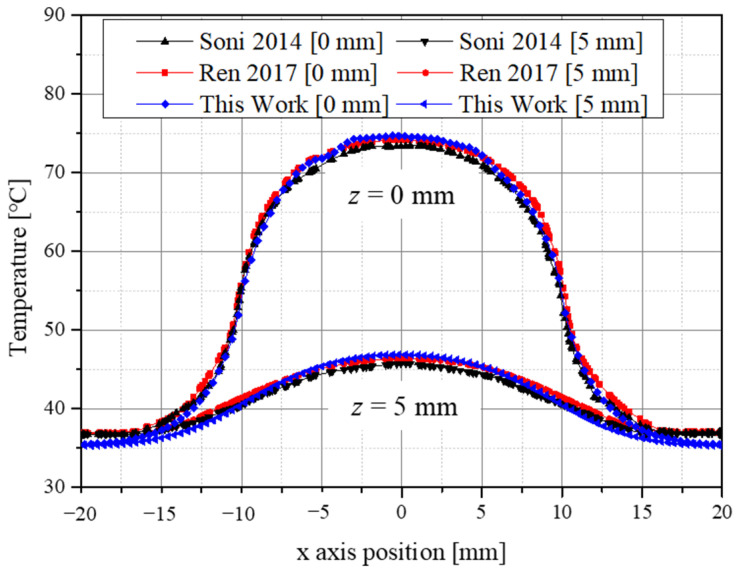
Validation result of the numerical model.

**Figure 3 sensors-22-01671-f003:**
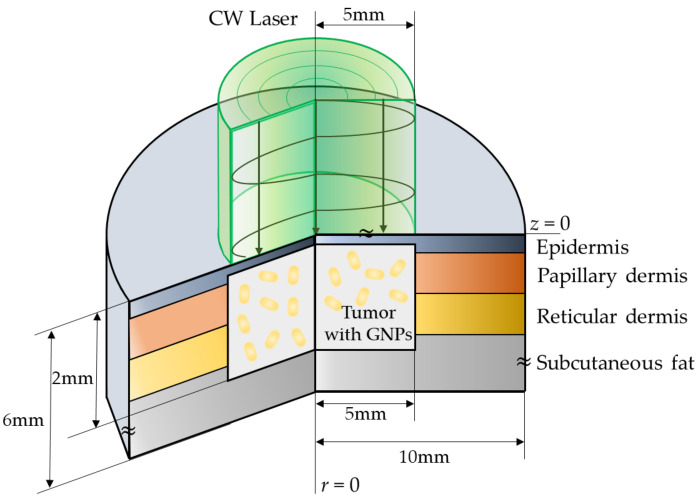
Schematic illustration of the numerical model.

**Figure 4 sensors-22-01671-f004:**
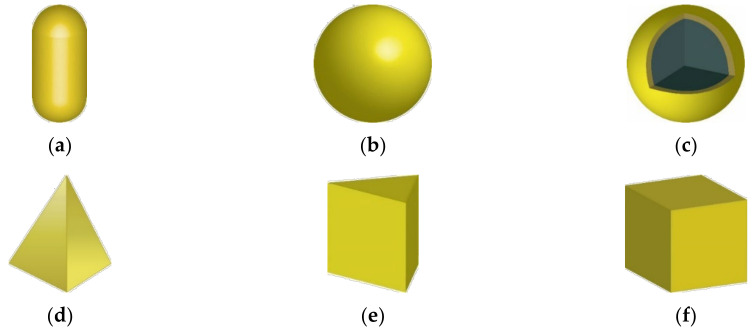
Various types of gold nanoparticles: (**a**) rod type; (**b**) sphere type; (**c**) shell type; (**d**) pyramid type; (**e**) prism type; and (**f**) cube type.

**Figure 5 sensors-22-01671-f005:**
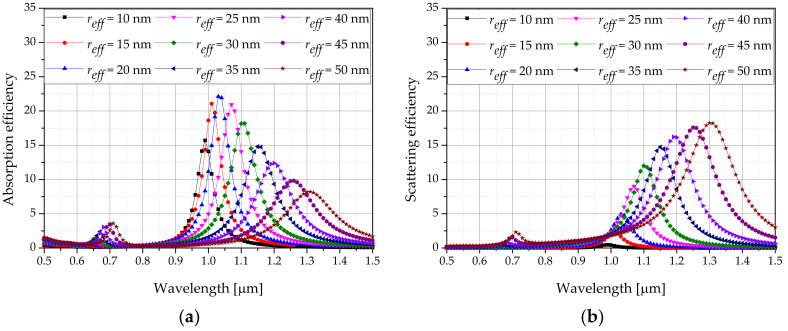
Optical efficiency of gold nanorods for various wavelengths: (**a**) absorption efficiency; (**b**) scattering efficiency.

**Figure 6 sensors-22-01671-f006:**
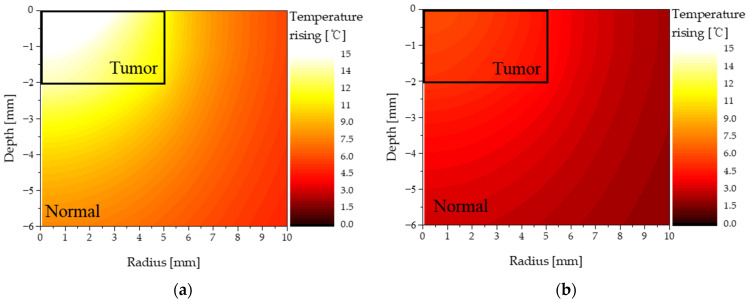
Temperature distribution of normal and tumor tissue (injected AuNPs: rod type): (**a**) $f_{v}$ = 10^−3^; (**b**) $f_{v}$ = 10^−6^.

**Figure 7 sensors-22-01671-f007:**
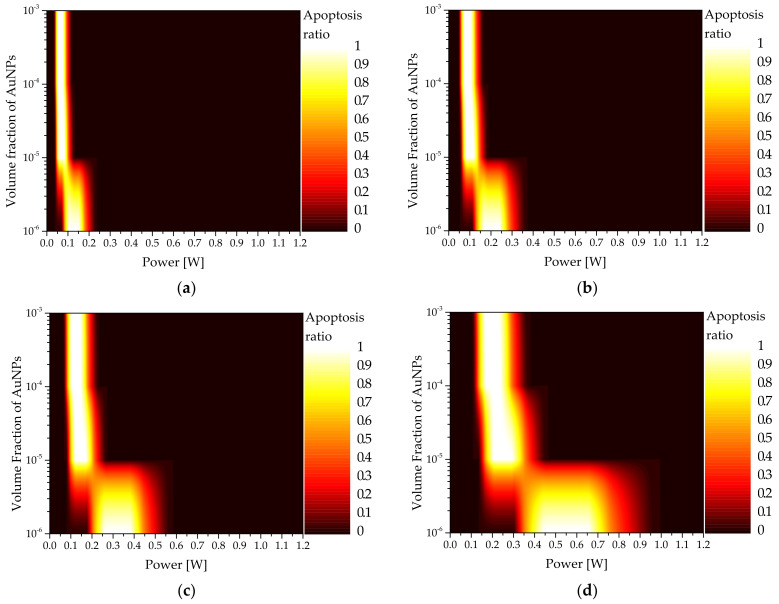
Apoptosis ratio ($\theta_{A}$) for various volume fractions of AuNPs ($f_{v}$ ) (injected AuNPs: rod type) (**a**) $r_{eff}$ = 10 nm; (**b**) $r_{eff}$ = 25 nm; (**c**) $r_{eff}$ = 35 nm; (**d**) $r_{eff}$ = 50 nm.

**Figure 8 sensors-22-01671-f008:**
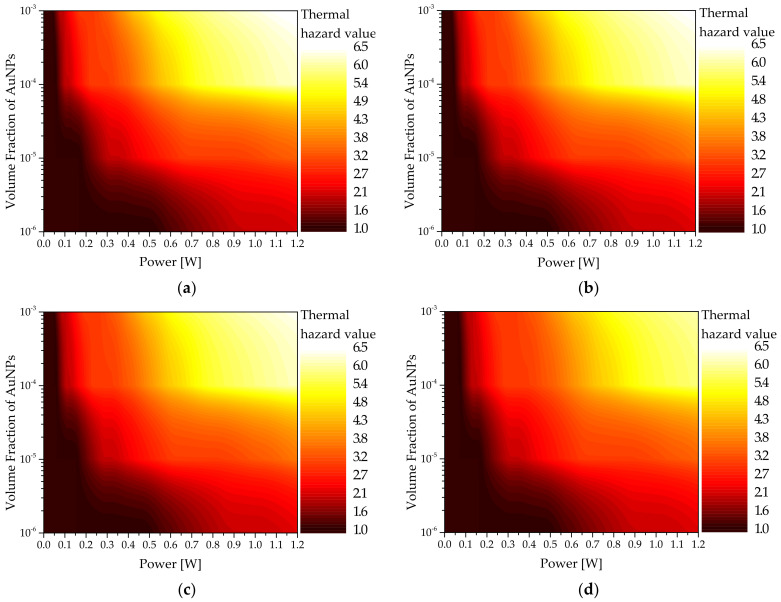
Thermal hazard value ($\theta_{H}$) for various volume fractions of AuNPs ($f_{v}$ ) (injected AuNPs: sphere type): (**a**) $r_{eff}$ = 10 nm; (**b**) $r_{eff}$ = 25 nm; (**c**) $r_{eff}$ = 35 nm; (**d**) $r_{eff}$ = 50 nm.

**Figure 9 sensors-22-01671-f009:**
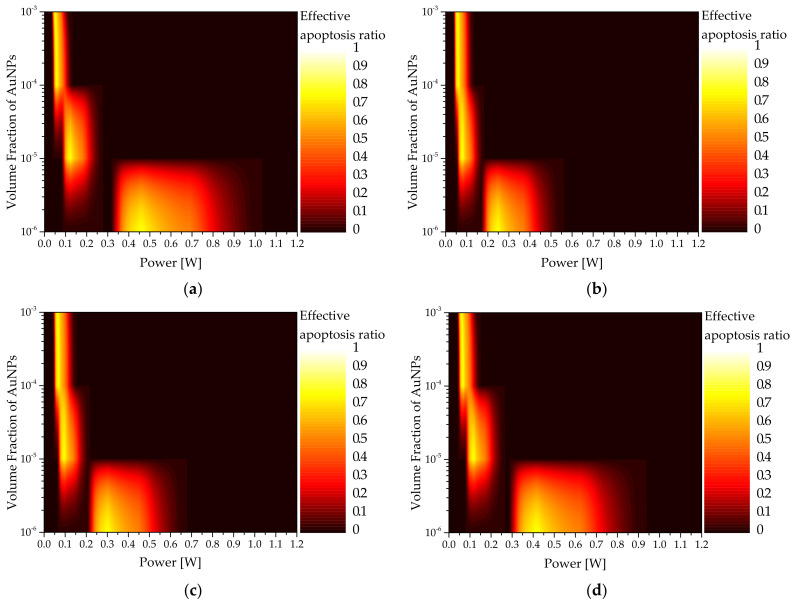
Effective apoptosis ratio ($\theta_{eff}$) for various volume fractions of AuNPs ($f_{v}$ ) (injected AuNPs: shell type): (**a**) $r_{eff}$ = 10 nm; (**b**) $r_{eff}$ = 25 nm; (**c**) $r_{eff}$ = 35 nm; (**d**) $r_{eff}$ = 50 nm.

**Figure 10 sensors-22-01671-f010:**
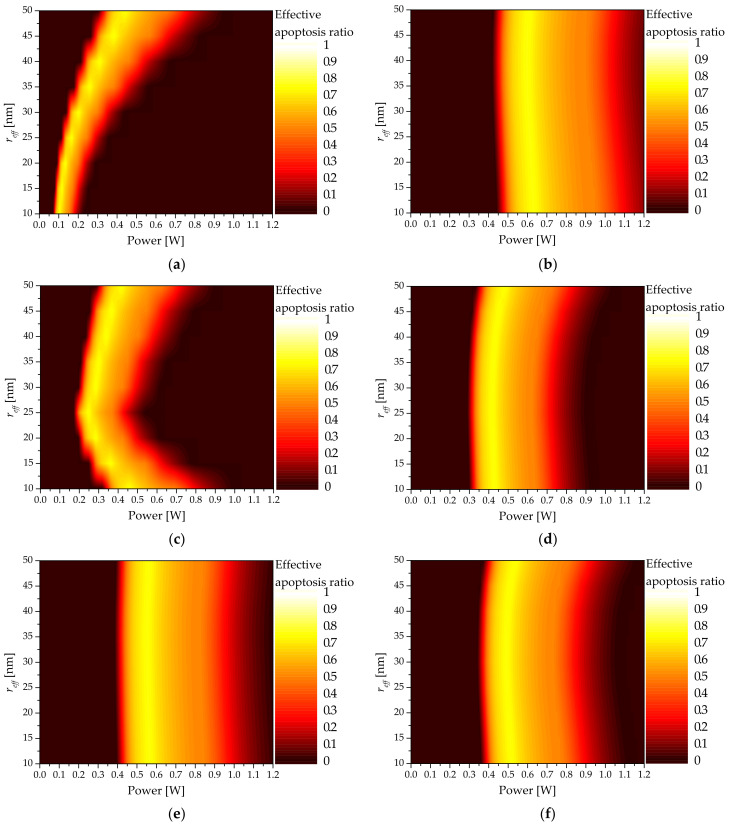
Effective apoptosis ratio ($\theta_{eff}$) for various types of AuNPs: (**a**) rod type; (**b**) sphere type; (**c**) shell type; (**d**) pyramid type; (**e**) prism type; (**f**) cube type.

**Table 1 sensors-22-01671-t001:** Laser-induced thermal effects [44,45].

Temperature Range (°C)	Biological Effect	Weight, w
37	Normal	1
37<T<43	Biostimulation	1
43≤T<45	Hyperthermia	2
45≤T<50	Reduction in enzyme activity	2
50≤T<70	Protein denaturation (coagulation)	3
70≤T<80	Welding	4
80≤T<100	Permeabilization of cell membranes	5
100≤T<150	Vaporization	6

**Table 2 sensors-22-01671-t002:** Thermal and optical properties used in the validation numerical analysis model.

	Tumor Tissue & AuNPs	Normal Tissue
Absorption coefficient (cm^−1)^	121	0.02
Reduced scattering coefficient (cm^−1^)	0.5	6.5
Density (kg/m^3^)	1100	1000
Specific heat (J/kgK)	4200	4200
Thermal conductivity (W/mK)	0.55	0.5

**Table 3 sensors-22-01671-t003:** Thermal properties of the skin layer and tumor [49,50,51,52,53,54].

	d (mm)	ρ (kg/m^3^)	$c_{p}(J/\mathrm{kgK})$	k (W/mK)
Epidermis	0.08	1200	3589	0.235
Papillary dermis	0.5	1200	3300	0.445
Reticular dermis	0.6	1200	3300	0.445
Subcutaneous fat	4.82	1000	2500	0.19
Tumor tissue	2	1070	3421	0.495

**Table 4 sensors-22-01671-t004:** Conditions used in the numerical analysis.

Numerical Parameter	Case	Number	Remarks
$\mathrm{Laser}\;\mathrm{power}\;(P_{l}$)	0 to 1.2 W	601	Interval: 0.002 W
Type of AuNP	Rod, sphere, shell, pyramid, prism, cube	6	
$\mathrm{Size}\;\mathrm{of}\;\mathrm{AuNP}\;(r_{eff}$)	10 to 50 nm	9	Interval: 5 nm
$\mathrm{Volume}\;\mathrm{fraction}\;\mathrm{of}\;\mathrm{AuNP}\;(f_{v}$)	10^−3^ to 10^−6^	4	Interval: 10^−1^

**Table 5 sensors-22-01671-t005:** Derivation of optical efficiency for AuNPs of various shapes.

AuNP Type	$r_{eff}(\mathrm{nm})$	$\lambda_{max}(\mathrm{nm})$	$Q_{abs}$	$Q_{sca}$
Rod	10.01	990	15.7160	0.4703
	15.58	1010	21.0660	2.2822
	20.02	1030	22.1100	4.9106
	25.58	1070	20.9410	9.0330
	30.02	1110	18.1880	11.7830
	35.59	1150	14.8070	14.7490
	40.03	1200	12.4000	16.1530
	45.60	1260	9.8333	17.5270
	50.04	1310	8.2106	18.2100
Sphere	10	510	0.4589	0.0016
	15	510	0.7113	0.0082
	20	510	0.9880	0.0266
	25	510	1.2914	0.0669
	30	510	1.6168	0.1424
	35	510	1.9491	0.2676
	40	510	2.2594	0.4531
	45	520	2.5419	0.8789
	50	520	2.7331	1.2601
Shell	9.96	530	1.1211	0.0040
	15.16	560	2.8056	0.0476
	20.11	590	5.6807	0.2735
	24.97	630	8.9640	1.0629
	30.03	650	8.6687	1.5817
	35.09	630	9.3365	2.5683
	39.99	860	8.4822	1.6203
	44.95	720	8.2782	1.7524
	49.98	720	7.1150	1.3602
Pyramid	10.04	590	1.3073	0.0044
	15.06	590	2.0064	0.0228
	20.07	600	2.7377	0.0804
	24.94	600	3.4829	0.1943
	29.96	610	4.1738	0.4310
	34.98	620	4.7951	0.8355
	40.15	630	5.2890	1.4857
	45.01	640	5.4828	2.3042
	49.88	650	5.3941	3.2793
Prism	10.02	530	0.6632	0.0017
	15.02	530	1.0137	0.0086
	20.03	530	1.3802	0.0268
	25.04	530	1.7562	0.0639
	30.05	540	2.1391	0.1424
	35.06	540	2.5227	0.2510
	40.06	540	2.8719	0.3962
	45.07	550	3.1703	0.6303
	50.08	550	3.4553	0.8469
Cube	9.93	530	0.8433	0.0031
	14.89	530	1.3110	0.0163
	19.85	530	1.8217	0.0532
	25.12	530	2.4007	0.1401
	30.09	530	2.9437	0.2905
	35.05	530	3.4159	0.5221
	40.01	540	3.8570	1.0409
	44.98	540	4.0785	1.5180
	49.94	540	4.0361	1.9763

**Table 6 sensors-22-01671-t006:** Optimal treatment conditions for various AuNP types and sizes.

AuNP Type	$r_{eff}(\mathrm{nm})$	$\lambda_{max}(\mathrm{nm})$	$f_{v}$	$P_{l}(W)$	Effective Apoptosis Ratio
Rod	10.01	990	10^−6^	0.104	0.74876
	15.58	1010	10^−6^	0.116	0.74978
	20.02	1030	10^−6^	0.134	0.75035
	25.58	1070	10^−6^	0.166	0.75046
	30.02	1110	10^−6^	0.202	0.75010
	35.59	1150	10^−6^	0.260	0.75058
	40.03	1200	10^−6^	0.310	0.75020
	45.60	1260	10^−6^	0.384	0.75054
	50.04	1310	10^−6^	0.442	0.75092
Sphere	10	510	10^−6^	0.630	0.75143
	15	510	10^−6^	0.626	0.75164
	20	510	10^−6^	0.620	0.75119
	25	510	10^−6^	0.612	0.75208
	30	510	10^−6^	0.606	0.75178
	35	510	10^−6^	0.598	0.75136
	40	510	10^−6^	0.598	0.75189
	45	520	10^−6^	0.600	0.75189
	50	520	10^−6^	0.606	0.75107
Shell	9.96	530	10^−6^	0.458	0.75111
	15.16	560	10^−6^	0.360	0.75065
	20.11	590	10^−6^	0.284	0.75089
	24.97	630	10^−6^	0.248	0.74977
	30.03	650	10^−6^	0.284	0.75083
	35.09	630	10^−6^	0.302	0.75042
	39.99	860	10^−6^	0.338	0.75095
	44.95	720	10^−6^	0.466	0.75094
	49.98	720	10^−6^	0.416	0.75081
Pyramid	10.04	590	10^−6^	0.430	0.75065
	15.06	590	10^−6^	0.426	0.75119
	20.07	600	10^−6^	0.420	0.75090
	24.94	600	10^−6^	0.418	0.75056
	29.96	610	10^−6^	0.418	0.75072
	34.98	620	10^−6^	0.424	0.75143
	40.15	630	10^−6^	0.434	0.75101
	45.01	640	10^−6^	0.454	0.75066
	49.88	650	10^−6^	0.480	0.75088
Prism	10.02	530	10^−6^	0.564	0.75141
	15.02	530	10^−6^	0.562	0.75164
	20.03	530	10^−6^	0.558	0.75132
	25.04	530	10^−6^	0.556	0.75146
	30.05	540	10^−6^	0.554	0.75149
	35.06	540	10^−6^	0.552	0.75140
	40.06	540	10^−6^	0.554	0.75114
	45.07	550	10^−6^	0.558	0.75140
	50.08	550	10^−6^	0.560	0.75141
Cube	9.93	530	10^−6^	0.518	0.75119
	14.89	530	10^−6^	0.510	0.75101
	19.85	530	10^−6^	0.502	0.75101
	25.12	530	10^−6^	0.494	0.75104
	30.09	530	10^−6^	0.490	0.75104
	35.05	530	10^−6^	0.492	0.75148
	40.01	540	10^−6^	0.496	0.75088
	44.98	540	10^−6^	0.512	0.75102
	49.94	540	10^−6^	0.534	0.75131

## Data Availability

Data sharing is not applicable to this article.

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
