# Peer review of "Numerical Study on Death of Squamous Cell Carcinoma Based on Various Shapes of Gold Nanoparticles Using Photothermal Therapy"

_sensors, 2022, doi:10.3390/s22041671_

Round 1
Reviewer 1 Report
Please see the attachment.

Author Response
Please confirm attached file.

Reviewer 2 Report
In this manuscript, the author reported numerical study based on heat transfer theory is conducted to investigate the death of squamous cell carcinoma located in the skin layer based on various shapes of gold nanoparticles (GNPs) used in photothermal therapy. This research work is innovative, but still some question needed to be answer. Given the current situation, I recommend its publication on Sensors after minor revisions.
- There were total seven paragraphs in the Introduction part. Every paragraph was too brief or ‘lacked depth’, it should be merged and enlarged to streamline the number of passages.
- Letters in most figures are too small to be identified. Authors are recommended to enlarge the legend letter.
- Conclusion part should be more concise.
- There are some typo errors in reference section. Authors need to revise the reference format accordingly.
- Related papers should be cited and discussed in the revised manuscript, e.g. Adv. Mater. 2020, 32, 1907855
Author Response
Please confirm attached file.

Reviewer 3 Report
The authors present a numerical study based on photothermal therapy using AuNPs applied to to skin tissues containing squamous cell carcinoma. The paper is interesting, although it is a numerical study, and the correspondent scientific literature presents a large number of similar studies, I believe that the medical community operating in the field of theranostics may find this study of great interest. Potentially, the authors could apply their findings to any cancer on which a photothermal therapy is handble.
However, I suggest to improve the study with mandatory modifications.
Severe changes: DDA is one of the most popular methods to simulate light scattering of arbitrarily shaped inhomogeneous nanoparticles, another method is the finite difference time domain method. These methods have a very similar region of applicability; however, they are rarely used together. I suggest to authors to justify why they are using only DDA and not compare their findings with other methods, for example finite difference domani method.
Medium changes: I consider table 4 the most important contribution of such paper, however, in theranostics application of photothermal therapy, the gold nanoparticles are functionalized with biomolecules that make able such nanoparticle to recognize cancer cells and not healthy cells. The question is how changes the optical response to laser excitation for gold nanoparticles as a function of dimensions and shapes?
what about the endocytosis processes to which the gold nanoparticle could be subjected?
On such two questions I suggest the author to cite:
Exploiting gold nanoparticles for diagnosis and caner treatments
D'Acunto et al, Nanotechnology, 2021 doi: 10.1088/1361-6528/abe1ed
Minor changes: the authors use the acronymous GNPs for gold nanoparticles, in literature is generally used AuNPs, where Au stay for aurum latin name of gold. I wish to ask to authors to prefer AuNPs to GNPs, but I leave the final decision on that to the authors.
I consider uncorrect the sentence (pag.2): " ...under various treatments conditions, such as GNP concentrations". The authors should defend this claim by citing papers on the subject, because the concentration of AuNPs is a key parameter of cancer treatments.
Author Response
Please confirm attached file.

Round 2
Reviewer 3 Report
The authors addressed all the questions proposed after first revision. I suggest that the paper as improved by the authors can be now published